# A Dedicated Veno-Venous Extracorporeal Membrane Oxygenation Unit during a Respiratory Pandemic: Lessons Learned from COVID-19 Part II: Clinical Management

**DOI:** 10.3390/membranes11050306

**Published:** 2021-04-21

**Authors:** Aakash Shah, Sagar Dave, Samuel Galvagno, Kristen George, Ashley R. Menne, Daniel J. Haase, Brian McCormick, Raymond Rector, Siamak Dahi, Ronson J. Madathil, Kristopher B. Deatrick, Mehrdad Ghoreishi, James S. Gammie, David J. Kaczorowski, Thomas M. Scalea, Jay Menaker, Daniel Herr, Ali Tabatabai, Eric Krause

**Affiliations:** 1Department of Surgery, Division of Cardiac Surgery, School of Medicine, University of Maryland, Baltimore, MD 21201, USA; sdahi@som.umaryland.edu (S.D.); rmadathil@som.umaryland.edu (R.J.M.); kdeatrick@som.umaryland.edu (K.B.D.); mghoreishi@som.umaryland.edu (M.G.); jgammie@som.umaryland.edu (J.S.G.); 2Program in Trauma, Department of Surgery, School of Medicine, University of Maryland, Baltimore, MD 21201, USA; sagar.dave@umm.edu (S.D.); kgeorge@umm.edu (K.G.); tscalea@som.umaryland.edu (T.M.S.); dherr@som.umaryland.edu (D.H.); 3Program in Trauma, Department of Anesthesiology, School of Medicine, University of Maryland, Baltimore, MD 21201, USA; sgalvagno@som.umaryland.edu; 4Program in Trauma, Department of Emergency Medicine, School of Medicine, University of Maryland, Baltimore, MD 21201, USA; amenne@som.umaryland.edu (A.R.M.); dhaase@som.umaryland.edu (D.J.H.); 5Perfusion Services, University of Maryland Medical Center, Baltimore, MD 21201, USA; bmccormi@umm.edu (B.M.); rrector@umm.edu (R.R.); 6Department of Cardiothoracic Surgery, University of Pittsburgh Medical Center, Pittsburgh, PA 15213, USA; kaczorowskidj2@upmc.edu; 7Department of Surgery, University of California San Francisco Medical Center, San Francisco, CA 94143, USA; jay.menaker@ucsf.edu; 8Program in Trauma, Department of Medicine, Division of Pulmonary and Critical Care, School of Medicine, University of Maryland, Baltimore, MD 21201, USA; atabatabai@som.umaryland.edu; 9Department of Surgery, Division of Thoracic Surgery, School of Medicine, University of Maryland, Baltimore, MD 21201, USA; ekrause@som.umaryland.edu

**Keywords:** extracorporeal membrane oxygenation, COVID-19, acute respiratory distress syndrome, tracheostomy, mechanical ventilation, pneumothorax, sedation, anticoagulation

## Abstract

(1) Background: COVID-19 acute respiratory distress syndrome (CARDS) has several distinctions from traditional acute respiratory distress syndrome (ARDS); however, patients with refractory respiratory failure may still benefit from veno-venous extracorporeal membrane oxygenation (VV-ECMO) support. We report our challenges caring for CARDS patients on VV-ECMO and alterations to traditional management strategies. (2) Methods: We conducted a retrospective review of our institutional strategies for managing patients with COVID-19 who required VV-ECMO in a dedicated airlock biocontainment unit (BCU), from March to June 2020. The data collected included the time course of admission, VV-ECMO run, ventilator length, hospital length of stay, and major events related to bleeding, such as pneumothorax and tracheostomy. The dispensation of sedation agents and trial therapies were obtained from institutional pharmacy tracking. A descriptive statistical analysis was performed. (3) Results: Forty COVID-19 patients on VV-ECMO were managed in the BCU during this period, from which 21 survived to discharge and 19 died. The criteria for ECMO initiation was altered for age, body mass index, and neurologic status/cardiac arrest. All cannulations were performed with a bedside ultrasound-guided percutaneous technique. Ventilator and ECMO management were routed in an ultra-lung protective approach, though varied based on clinical setting and provider experience. There was a high incidence of pneumothorax (n = 19). Thirty patients had bedside percutaneous tracheostomy, with more procedural-related bleeding complications than expected. A higher use of sedation was noted. The timing of decannulation was also altered, given the system constraints. A variety of trial therapies were utilized, and their effectiveness is yet to be determined. (4) Conclusions: Even in a high-volume ECMO center, there are challenges in caring for an expanded capacity of patients during a viral respiratory pandemic. Though institutional resources and expertise may vary, it is paramount to proceed with insightful planning, the recognition of challenges, and the dynamic application of lessons learned when facing a surge of critically ill patients.

## 1. Introduction

Acute respiratory distress syndrome (ARDS) is a form of severe respiratory failure that is associated with a mortality as high as 45% [1]. The use of veno-venous extracorporeal membrane oxygenation (VV-ECMO) has become a mainstay of therapy for refractory ARDS [2,3,4,5,6,7]. Our center has demonstrated that having a standardized approach with a dedicated lung rescue unit (LRU) can improve outcomes [8].

While the COVID-19 disease, caused by the novel coronavirus 2019 (SARS-CoV2), can cause acute respiratory failure, it has become clear that there are distinctions from traditional ARDS. The vasocentric features of the pulmonary insult in COVID-19 lead to a unique lung injury–COVID acute respiratory distress syndrome (CARDS) [9]. Early autopsy studies demonstrated vascular congestion, pneumocyte necrosis, and diffuse alveolar disease [10]. It has been proposed that the alveolar viral damage may lead to an inflammatory reaction and microvascular pulmonary thrombosis, with a resultant progressive endothelial thromboinflammatory syndrome, and subsequent multi-organ failure [11]. Similar to ARDS, CARDS carries a high mortality with reports ranging from 26–61.5% [12,13,14]. The VV-ECMO provided by experienced centers may play a role in CARDS refractory to ventilator and medical management [15,16].

Here, we present our practice preferences, challenges faced in the initial surge of COVID-19 patients, and the adjustment in care that was made. In the current paucity of a high level of evidence, and while awaiting large studies to direct the management of these patients, we present a descriptive analysis of our institution’s trials and tribulations through the first surge of COVID-19 patients. These practices are not applicable at every center, yet we hope this may guide institutions facing similar obstacles and help plan for the subsequent surges of patients during this respiratory viral pandemic. We discuss our experience, and the lessons we learned along the way, in the management of our first 40 CARDS patients requiring VV-ECMO in a dedicated biocontainment unit (BCU) [17], from which 21 survived to discharge and 19 died.

## 2. Materials and Methods

We conducted a retrospective review of patients that required VV-ECMO, from March to June 2020, at our institution. Forty-three patients were identified, from which 40 patients had COVID-19 and were included in the study. We reviewed our institutional strategies for managing those patients within the BCU. An institutional review board approval was obtained and the need for consent was waived for this study (HP-00090914). The data collected included the time course of admission, VV-ECMO run, ventilator length, hospital length of stay, and major events related to bleeding, such as pneumothoraces and tracheostomy. These data were obtained through the review of patient charts in the electronic medical record system. The dispensation of sedation agents and trial therapies were obtained from institutional pharmacy tracking. The care of the patients in the BCU required a multi-disciplinary approach, including intensivists, surgeons, and other specialists when indicated, and the approach to the varying aspects of management for these patients were made as a group. A descriptive statistical analysis was performed. Continuous variables were evaluated for normalcy and reported as a mean with standard deviation or a median and interquartile range, as appropriate. Categorical variables were reported along with the numbers and percentages.

## 3. Results

### 3.1. ECMO Consultation Process and Indications for Support

Prior to the COVID-19 pandemic, we established a well-organized process for ECMO consultation in which consults were processed through a central call center, allowing for a single conference call with key members. This model continued through the COVID-19 pandemic, with a team that consisted of a cardiac surgeon, a biocontainment unit (BCU) intensivist, and a critical care resuscitation unit intensivist [18]. The acceptance of patients with CARDS for ECMO consideration to our institution depends on consensus from the team, with a central intensivist physician serving as an adjudicator for resource allocation throughout the entire hospital system [17].

Our ECMO program has established indications and contraindications for VV-ECMO [8]. While the criteria for ECMO initiation did not significantly change during the pandemic, there were additional adjustments made for COVID-19 patients, based on institutional and state-level discussions (Table 1). These changes included age, body mass index (BMI), and neurologic status/cardiac arrest. The lower age cutoff at 55 years was implemented due to a rapidly increasing mortality rate above this age in COVID-19 cases [19,20]. While obesity itself has not typically been a contraindication to VV-ECMO support [21,22], the mortality associated with obesity in COVID-19 patients [23,24] led us to use a BMI > 40 kg/m^2^ as a relative contraindication. Previously, patients were placed on VV-ECMO support in the cases of unknown neurologic status after, or even during, a cardiac arrest in the setting of presumed hypoxia [25]; however, we viewed this as a strong relative contraindication for COVID-19 patients.

As a program, we made the decision to limit the use of extracorporeal cardiopulmonary resuscitation (ECPR) during the COVID-19 pandemic to select situations, such as an extension of post-cardiotomy care. The need for a rapid cannulation and restoration of systemic support is paramount to patients in cardiac arrest [26]. We believed that the time needed to appropriately don full personal protective equipment (PPE) and to set up equipment in an isolation room would result in a significant delay. Furthermore, cannulating without appropriate PPE would be an unacceptable risk for the ECMO team. The benefit-to-risk ratio of the COVID population, at the time of the initial surge, led to our decision not to offer ECPR. While our program does not offer mobile ECMO, patients already on VV-ECMO were admitted [27].

All of our indications and contraindications were relative. This allowed for bedside provider discretion when making the final decision. As a result, some patients were supported with VV-ECMO, which fell outside of our guidelines (Table 2). Ten cannulated patients had a BMI > 40 kg/m^2^, from which four died, and six decannulated. Four patients had suffered cardiac arrest prior to cannulation, from which two had a GCS of 8 and 11. Of these four patients, three died and one was decannulated and discharged. Three additional patients were cannulated with a best GCS of 3 and no prior cardiac arrest, and all three survived to hospital discharge. All of these patients were discussed extensively prior to cannulation, and there was a consensus approval to proceed with cannulation. As the pandemic continued, resources became scarce our experience grew, and there was a stricter adherence to the established criteria. The decision to proceed with cannulation in our program is always a multi-disciplinary discussion. The burden of withholding ECMO, a potentially lifesaving support, should never fall on one’s shoulders alone.

Lessons:Establishing consensus criteria for COVID-19 ECMO support based on institutional and/or state-level discussion.The use of a multidisciplinary ECMO team is beneficial, particularly when considering variations from criteria on a case-by-case basis.Considering the impact of exceptions to criteria on resources during a pandemic.The safety of the ECMO team is paramount during cannulations for COVID-19.Continuing to revise VV-ECMO criteria for COVID-19 patients, as outcome data accumulate.

### 3.2. ECMO Initiation

Our standard approach is a bedside, percutaneous technique, with separate peripheral drainage and return cannulae. Vascular access was obtained with ultrasound-guidance and transthoracic echocardiography, used to position the femoral cannula at the level of the inferior cavo-atrial junction [28]. The team in the room consisted of six key members—two cardiac surgeons, an intensivist, a nurse, a respiratory therapist, and a perfusionist. We safely instituted VV-ECMO in COVID-19 patients in the BCU, with no immediate complications in all 40 patients.

With the reports of severe hypoxia and ventilation-perfusion (VQ) mismatch in these patients, along with the logistical difficulties of upsizing cannulae in isolation rooms, larger cannulae were placed with the goal of achieving higher indexed flows. The preferred cannulae were a 25F venous HLS cannula (Getinge, Wayne, NJ, USA) in the common femoral vein, for drainage, and a 23F arterial HLS cannula (Getinge) in the right internal jugular (RIJ) vein, for return (n = 34). No dual-lumen cannulae were used, as this is not our institution’s preference. Reports from Italy, New York, and China indicated a subset of patients who developed myocarditis or cardiac failure secondary to pulmonary insult [29,30]. Based on these early publications, in patients on high doses of vasoactive, a femoral arterial sheath was placed at the time of VV-ECMO initiation, with the anticipation of potential conversion to veno-arterial (VA) or hybrid (veno-arterio-venous) ECMO support. However, this practice was eventually eliminated as part of our initial cannulation process, as only one patient required conversion to hybrid ECMO support due to cardiac arrhythmias and subsequent hemodynamic instability peri-cannulation. This patient did not survive.

Lessons:An established institutional practice with a dedicated team for ECMO initiation in patients with CARDS should be continued, whether in the operating room or with fluoroscopy; however, if unable to transport patients, bedside percutaneous cannulation can be safely performed.For bedside cannulation, a consistent, well-defined team of people to manage varying aspects of patient care is essential for patient safety, particularly while in an airlock.Our experience rarely required VA-ECMO support for CARDS: This may be different for other centers, and a multidisciplinary approach to patient selection and initiation should be utilized in these cases.

### 3.3. Right Ventricular Strain

Prior to the COVID-19 pandemic, in patients requiring VV-ECMO, low-dose epinephrine and inhaled epoprostenol were empirically administered and subsequently weaned after the confirmation of normal right ventricular (RV) function. Though benefit is suspected with inhaled pulmonary vasodilators for COVID-19 patients [31,32,33,34,35], early in the BCU, the decision was made to limit the epoprostenol administration. This decision was multifactorial, including cost, availability, and use of a continuous aerosolizing substance leading to potential exposure to the care team. In addition, epoprostenol led to a frequent turnover of ventilator filters and altered the flow readings on the ventilators. As a result, inhaled epoprostenol and inotropic support were reserved for patients with evidence of echocardiographic RV strain. During the initial surge of the pandemic, we were unable to perform our protocol of serial formal echocardiography by an ultrasound technician, due to efforts to optimize the use of limited PPE and exposure in the airlock of the BCU. This led to further training and a complete dependence on bedside sonography by intensivists for assessment on RV function in order to make decisions on inotropic support and volume management.

Lessons:Consider the empiric use of inhaled pulmonary vasodilators and inotropic support for RV strain in the setting of ECMO-dependent respiratory failure, with subsequent weaning.Patients on VV-ECMO in an airlock unit require vigilant monitoring. Bedside sonography can be very helpful for the re-evaluation of cardiac function; specifically, in a setting with limited PPE.The use of inhaled vasodilators may have logistic disadvantages that limit their use.

### 3.4. Ventilator and ECMO Circuit Management

Traditionally, ARDS is treated with lower tidal volumes and higher positive end-expiratory pressure (PEEP) [36,37,38,39]. With a focus on preventing ventilator-induced lung injury (VILI) and patient self-inflicted lung injury (P-SILI), the previous LRU standard approach to ventilator management for patients on VV-ECMO had been a lung-protective strategy with pressure control (PC) [8,38,40,41,42,43]. This was not always feasible in patients with COVID-19.

Unlike ARDS, CARDS patients face a highly abnormal coagulation cascade, leading to pulmonary coagulopathy and a high VQ mismatch [44,45,46]. The presentation of CARDS falls on a spectrum between the extremes of 2 phenotypes: type L (low elastance and high compliance) and type H (high elastance and low compliance) [47]. Many patients, prior to admission to the BCU, were described to have good lung compliance, poor oxygenation, and elevated minute ventilation, which were similar to previous reports [12,48]. A majority of the patients transferred to the BCU had respiratory failure that was refractory to standard ARDS protocols. Some patients had a trial of personalized airway pressure release ventilation (APRV), either prior to admission or upon admission to the BCU [49,50,51]. At times, the initial presentation would be type L, which, over the hospital course, would develop into type H, potentially due to a proposed mechanism of persistent vasocentric hypoxemia, increased edema, and P-SILI [9].

Once cannulated, a majority of patients were placed on lung-protective settings. Select patients were on APRV at the time of cannulation and remained on this mode while being transitioned to lung-protective settings. This was done by keeping the T_high_ at the desired mean airway pressure. PEEP on APRV is calculated by utilizing an expiratory hold. Patients had ventilator adjustments based on clinical status to aim for a plateau pressure below 30 cm H_2_O and driving pressure of less than 15 cm H_2_O, as previously described [39,50]. If the patient was persistently hypoxic or hypercapnic, the flow or sweep was increased, respectively. In the presence of a compensatory metabolic alkalosis, a PaCO_2_ correction should be done cautiously with the initiation of VV-ECMO. A rapid correction can lead to a systemic alkalosis and potential neurologic insult [52,53].

Compared to our previous experience in the LRU, despite lung-protective ventilation strategies and maximal ECMO support, there was a higher incidence of refractory hypoxemia and hypercapnia. Prone positioning was implemented if patients had improvement with prone positioning prior to ECMO support, radiographic evidence of West Zone III injury, or refractory hypoxia. Consistent with the reports of improved oxygenation with prone positioning in ventilated patients with CARDS, we noted a similar trend in our ECMO-supported patients [54]. Other implemented options for refractory hypoxia included targeted temperature management, neuromuscular blockade, higher hemoglobin goals, and beta blockade [7,55,56,57,58]. For refractory hypercapnia, a parallel oxygenator with an in-line blender, or dual blenders, were added to the circuit to increase the sweep flow (n = 3), though these patients ultimately did not survive.

Through the COVID-19 pandemic, in the BCU, there was a great deal of variance in management for VV-ECMO patients. This stemmed from the differences in both philosophical and physiological approaches among intensivists. The pandemic led to a need for more intensivists which ultimately led to varying levels of experience with caring for VV-ECMO patients and different ventilator modes [17]. At times, this led to an increased ventilator support with the risk of VILI, in premature attempts to wean from VV-ECMO support. When the transition to dedicated ECMO and non-ECMO attendings was made, this variance decreased, and a more standardized approach was adopted. Though it is currently unclear what the best approach is for patients with CARDS requiring VV-ECMO, Figure 1 shows our proposed algorithm for ventilator and circuit management, as well as options for refractory states.

Lessons:CARDS has distinct pathophysiologic differences to ARDS, and the previous standard ARDS approaches may not be entirely applicable.Once lung-protective settings are achieved, consider adjustments to ECMO settings prior to increasing the ventilator support, due to the dangers of VILI.It is important for intensivists to have a high level of experience with ECMO in the management of ventilated patients on VV-ECMO support during a respiratory viral pandemic such as COVID-19.Though the optimal strategy for the management of these patients is unclear, a standardized algorithmic approach to ventilator and ECMO circuit management in CARDS patients may increase safety.

### 3.5. Pneumothoraces

Pneumothorax is a known complication, with associated poor prognosis in patients with ARDS [59]. There is a paucity of data on the incidence in patients supported with VV-ECMO; however, in our experience, pneumothorax is an infrequent complication when ECMO is used in concert with a lung-protective ventilatory strategy. Though the incidence of pneumothorax seems to be low in COVID-19 patients in the current literature, we noted a relatively high incidence (n = 19) in patients who required VV-ECMO support for CARDS [60,61,62].

Ultrasound has been established for bedside diagnosis for pneumothoraces, which was further underscored by our experience in the BCU, where it was readily available in an airlock, when compared to X-ray. After diagnosis, if a patient suffers a pneumothorax and yet is hemodynamically stable, heparin is held for 4 h before and after chest tube placement. Ultrasound can be useful to guide the placement of chest tubes. Our institutional preference for intervention on pneumothoraces in VV-ECMO-supported patients has been the placement of an 8F pericardiocentesis catheter (Cook Medical, Bloomington, IN) by an experienced provider [63,64,65].

We found that nearly half of the COVID-19 patients had persistent pneumothoraces, of unclear clinical significance, despite the 8F catheters which led to multiple repositioning, upsizing to a 14F Wayne pneumothorax catheter (Cook Medical), or the placement of an additional drain. Since many catheters were kinked and obstructed, one potential contributing factor might be the strain of being in an airlock, which led to less frequent bedside assessments, making routine chest tube care difficult to maintain.

Given these experiences, our management of pneumothoraces was modified. The size, location, timing, and significance of the pneumothorax was determined. If the pneumothorax is new-onset, an 8F catheter is placed at the bedside. Even if the follow-up chest X-ray shows that the lung is not fully expanded despite −20 mm Hg of suction, no further intervention is required, as long as the tube is patent and the patient is stable. If there is a significant air leak that a small caliber chest tube cannot fully capture, a stable patient is taken for a chest computed tomography (CT) scan to be evaluated for loculations, adhesions, or evidence of trapped lung. In an unstable patient, the catheter is upsized over a guidewire to a 14F catheter. If the CT scan demonstrates a large loculation, which is clearly not a pneumatocele, then the placement of an image-guided catheter is considered. Subsequently, if the lung does not fully re-expand and appears trapped, further interventions are likely not warranted until the underlying lung function improves. As many of these patients have prolonged air leaks, we do not remove the pleural drains until they have been liberated from the ventilator. At that point, we perform a clamp trial (24–48 h) before removing the tube.

Of the 19 patients who required treatment of their pneumothorax, 14 died and 5 were discharged (Figure 2). Two patients required emergent thoracotomies for bleeding after the placement of pleural catheters, who subsequently died. Care must be taken to avoid areas of adhesion, as the insertion of tubes in these areas may result in injury. Lung re-expansion in areas of tethering may also result in injuries. The high incidence of pneumothorax seen may be representative of a subset of patients with more severe lung disease or associated with a longer duration of mechanical ventilation due to VV-ECMO support. Our experience suggests that the development of a pneumothorax in COVID-19 patients on VV-ECMO is a poor prognostic sign; moreover, any major procedural complications from chest tube or pleural catheter placement may not be tolerated.

Lessons:Experienced providers should place chest tube and pleural catheters, as complications may not be tolerated in patients with CARDS on VV-ECMO.Routine chest tube care may be sufficient to prevent radiographic recurrence of pneumothoraces.The pleural space may be complex, and it is important to be mindful of lung adhesions/tethering, despite the acute nature of presentation, when deciding the entry point for chest tube placement (CT imaging, if safe to perform, is preferable).A persistent pneumothorax on imaging, despite multiple interventions, may be indicative of a trapped lung, and further intervention is likely not warranted until the underlying lung function improves for patients with COVID-19.

### 3.6. Tracheostomy

There is a multitude of theories and discussions about the proper timing, location, procedure, and management of tracheostomies in COVID-19 patients [66,67]. Initially, our institutional preference was not to perform tracheostomies on COVID-19 patients. It was quickly recognized that tracheostomies could be safely performed, and early tracheostomies were implemented within four to six days of cannulation [68]. During this period, there was a higher bleeding complication rate than we had previously experienced. Additionally, many patients did not survive long after their tracheostomies, due to the course of their disease. We then modified our practice to wait approximately 10–14 days on ECMO, until the patients proved to have longer periods of hemodynamic stability, minimal ventilator support, and no significant coagulopathy (Figure 2).

We performed percutaneous tracheostomies (n = 32) at the bedside with a similar team for VV-ECMO cannulation. Initially, in order to prevent aerosolization, all tracheostomies were performed on apneic patients, relying on VV-ECMO support during the procedure. We quickly modified this after nearly all the patients had significant setbacks in their tidal volumes post procedure. This was likely due to a rapid collapse of the poorly distensible lung parenchyma; several patients developed pneumothoraces within 72 h of their tracheostomy (n = 5), which was thought to be due to increased barotrauma from recruitment attempts. To mitigate this, we now continue ventilation while performing the entire procedure, though it is yet to be determined if this improves clinical outcomes.

While tracheostomy site bleeding is a known, common complication when performed on VV-ECMO, we experienced an unusually high incidence of bleeding requiring interventions in COVID-19 patients (n = 5). Two patients required transfusion and packing of the tracheostomy tract, and 3 required tracheostomy revisions. While the underlying coagulopathy, as seen in COVID-19 patients, and the anticoagulation for VV-ECMO contributed to some bleeding, pressure necrosis can further the complications. Despite the strain on hospital resources and personnel, tracheostomy care and maintenance is essential.

Our approach to tracheostomy is in contradiction to at least one other report. Mustafa et al., utilized a dual-lumen cannulation approach with an early extubation strategy for 40 patients [69]. Hence, the optimal strategy for early, delayed, or no tracheostomy (early extubation) remains to be determined for CARDS patients.

Lessons:The optimal timing for tracheostomy in COVID-19 patients on VV-ECMO has yet to be determined; however, consider waiting until patients have proven to have an extended period of hemodynamic and respiratory stability, and absence of a significant coagulopathy.Vigilance to routine tracheostomy care may prevent, or decrease, the incidence of bleeding complications in patients with CARDS.

### 3.7. Anticoagulation

Heparin has traditionally been used in the LRU for anticoagulation and titrated to a goal partial thromboplastin time (PTT) of 45–55 s. In the BCU, COVID-19 patients were managed similarly, though early on they were noted to have an increased incidence of thrombotic complications, such as oxygenator clotting. Given these repeated observations and other reports of prothrombotic events, we adjusted our PTT goal to 60–80 s [70,71,72]. Anticoagulation dosing for patients on continuous renal replacement therapy and general venous thromboembolism prophylaxis was also higher (Figure 3).

Patients on ECMO may develop acquired coagulopathies from exposure of blood to artificial surfaces and high shear stresses [72,73]. This was prevalent in our patient population. Acquired von Willebrand deficiency was observed and treated with the replacement of factors [73]. Patients with heparin-resistance due to anti-thrombin III deficiency (n = 1) and suspected (n = 4), or those with confirmed heparin-induced thrombocytopenia (n = 3), were transitioned to a direct thrombin inhibitor with a goal PTT of 46–76 s.

The management of a potential prothrombotic state, with a subsequent coagulopathic dysregulation as a sequela of disease state in COVID-19, proved to be challenging to manage [74,75]. These patients had a higher rate of procedural-related bleeding complications and oropharyngeal mucosal bleeding, with two patients requiring interventional radiology embolization. Ultimately, in select patients with significant bleeding, anticoagulation was held, and flows > 4 L/min were maintained in order to minimize potential thrombotic complications within the ECMO circuit [76]. Despite this, if bleeding was persistent, patients were considered for early decannulation, depending on their respiratory status.

Lessons:Consider a higher baseline anticoagulation therapeutic goal than that traditionally used in VV-ECMO, given the pro-thrombotic state seen in COVID-19.Be aware of the potential for the development of coagulopathy on VV-ECMO in CARDS, and the importance of an algorithmic approach for anticoagulation.

### 3.8. Sedation

Sedation and pain management were a particularly difficult challenge for this patient population, due to the disease process, system setup, and scarcity of resources. Similar to other viral infections, COVID-19 patients displayed neurotrophism with signs of tachypnea and agitation not rooted in specific physiologic distress [77,78,79]. This led to an increased use of sedation, particularly with patients on ECMO, as ventilator dyssynchrony led to P-SILI and flow variability led to hemodynamic instability.

Up-titration of sedation was used to mitigate some of the risk of physical harm to the patients and/or care team as a sequela of agitation [17]. Moreover, due to the rotating schedule of staff in the airlock, medications were often scheduled together, instead of in a staggered fashion, to minimize the strain on pharmacy and nursing teams. This, at times, led to significant fluctuations in hemodynamics because of the concurrent administration of multiple sedating agents. Sedation was prioritized with a vigilant monitoring of QTc, especially when antipsychotics were co-administered with antimicrobials. In addition, there was a toleration of higher triglyceride levels with adjusted nutrition in the setting of propofol use.

Lastly, due to the increased utilization and decreased availability of sedation medications, as seen throughout the country [80], alternatives for preferred sedation regimens were utilized. Guidelines for the use of various agents were disseminated among the care teams. There was a period when fentanyl was on shortage and hydromorphone use was increased. The increase in opioid use as a sedation agent was also seen. After this was adjudicated and fentanyl became available, hydromorphone and overall opioid use were decreased (Figure 4).

Lessons:Team nursing may have the consequent risk of an increased use of sedation for feasibility and safety for patients.Co-administration of multiple medications can ease the strain on pharmacy and nursing, but this must be counterbalanced with hemodynamic considerations.Availability of multiple pharmacologic regimens is important when faced with drug shortages.

### 3.9. Indication for Decannulation

Prior to COVID-19, patients in the LRU were evaluated for decannulation with a no-sweep trial, when they were on minimal ventilatory support (i.e., pressure support ≤ 15 cm H_2_O, PEEP ≤ 10 cm H_2_O), and minimal inspired oxygen concentration. During COVID-19, these trials were started when ventilatory support was higher than that in the LRU, in order to expedite patients for decannulation, given the limited resources of ECMO and the number of patients requiring support. Additionally, the coagulopathy and bleeding complications observed in COVID-19 patients with prolonged ECMO [81,82,83,84] led the team to be more aggressive with decannulation.

In the LRU, during the pre-COVID conditions, most patients had to tolerate no-sweep for 24 h continuously, in order to be decannulated. Patients who were on ECMO for extended periods of time or were centrally cannulated required longer periods of no-sweep prior to decannulation. During the COVID-19 pandemic, the practice in the BCU was to have patients off sweep for a minimum of 48–72 h, continuously. The decision was initially made to minimize any emergent re-cannulation in the BCU, which is significantly more challenging in an airlock space. As our experience grew, we observed that many patients failed these trials between 24–48 h, thus solidifying our decision of extended trials. Furthermore, once the patients were successfully decannulated from VV-ECMO for 24 h, they were transferred out of the BCU to other COVID-19 units, allowing an available bed for other patients that may require VV-ECMO support. No patient required re-cannulation using this approach.

Lessons:With a surge of patients who may require ECMO support, patients may need to be decannulated from ECMO at higher than traditionally accepted levels of ventilatory support.Consider the feasibility and logistical challenges of re-cannulation in this setting when determining the length of the no-sweep trial prior to decannulation.In patients with persistent, refractory coagulopathy, decannulation may need to be performed earlier with concurrent increases in ventilatory support, if feasible.

### 3.10. Trial Therapies

In the BCU, many patients were treated with trial therapies for COVID-19 [85,86]. Initially, most patients were treated with hydroxychloroquine and zinc; however, as data became available, this practice was no longer utilized [87,88]. Other trial therapies included, but were not limited to, remdesivir, tocilizumab, stem cells, and convalescent serum. One of the mainstays of treatment in the BCU was steroids. If a patient was in septic shock and on multiple vasoactives, stress dose steroids were used [89,90,91,92,93,94]. If a patient began to develop signs of overt inflammatory response, a high dose of dexamethasone over 10 days was used [93]. If poor lung compliance was noted, a 28-day methylprednisolone protocol was used [95,96]. Whether these therapies were effective is yet to be determined.

Lessons:It is important to keep up to date with the changing landscape of literature for various trial therapies and to consider implementation with caution.

## 4. Discussion

The management of patients with CARDS is challenging, particularly in those patients with refractory disease that require VV-ECMO support. The care of these patients requires multidisciplinary expertise, particularly in the setting of a surge of patients during a pandemic. The early establishment of consensus criteria and guidelines can provide institutional consistency for the implementation of VV-ECMO in CARDS, and any deviation from this should be discussed as a team on a case-by-case basis.

Though staff may need to be reassigned to care for these patients and accommodate for the increased capacity, it is important to have a multidisciplinary team—surgeons, intensivists, pharmacists, nutritionists, and infectious disease physicians—with ECMO expertise to guide their care. Previously-established institutional practices and preferences should be maintained, although some deviations may be required due to increased patient-to-staff ratios, challenges of an airlock unit, and resource scarcity. These modifications need to be implemented in a well-organized and structured manner.

We found a few key differences in our management of patients with CARDS, compared to traditional ARDS. Ventilator management for CARDS may require an algorithmic approach in concert with changes to ECMO settings, guided to specific classes of physiology. The prothrombotic state in patients with COVID-19 and the potential subsequent coagulopathy from ECMO can be challenging to manage. Persistent bleeding may require the continuation of VV-ECMO without anticoagulation, or, an ultimately earlier decannulation. In the setting where patient-to-staff ratios are increased and the staff is managing multiple critically ill patients, it is important to maintain routine tasks, such as chest tube and tracheostomy care. A checklist approach may prove useful.

As detailed through our experience in a high-volume ECMO center, there are challenges that come with taking care of an expanded capacity of patients during a viral respiratory pandemic. Our preferences and practice may not apply to all hospitals, and we implore centers caring for patients with COVID-19 requiring extracorporeal life support to consider reflecting on their challenges through this pandemic. Though institutional resources and expertise may vary, it is paramount to proceed with insightful planning, the recognition of challenges, and a dynamic application of the lessons learned when facing a surge of critically ill patients.

## Figures and Tables

**Figure 1 membranes-11-00306-f001:**
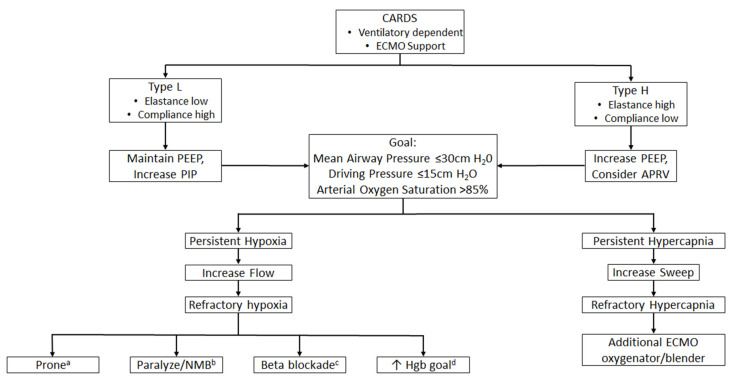
Ventilator algorithm for patients with COVID-19 acute respiratory. ^a^ proning: if radiographic evidence supports benefit, or if patient previously benefited from proning, prior to ECMO support. ^b^ paralysis or neuromuscular blockade: if vent dyssynchrony occurs, or if transpulmonary pressure is high due to native breaths. ^c^ beta blockade: if heart rate is above 80 beats per minute, it requires vigilant monitoring for end-organ dysfunction (e.g., lab tests such as lactate) and frequent assessments of cardiac function (i.e., bedside echocardiography). ^d^ increase hemoglobin goal: if hemoglobin is less than 9 g/dL. Legend: APRV—airway pressure release ventilation; CARDS—COVID-19 acute respiratory distress syndrome; ECMO—extracorporeal membrane oxygenation; Hgb—hemoglobin; HR—heart rate; NMB—neuromuscular blockade; PEEP—positive end expiratory pressure; and PIP—peak inspiratory pressure.

**Figure 2 membranes-11-00306-f002:**
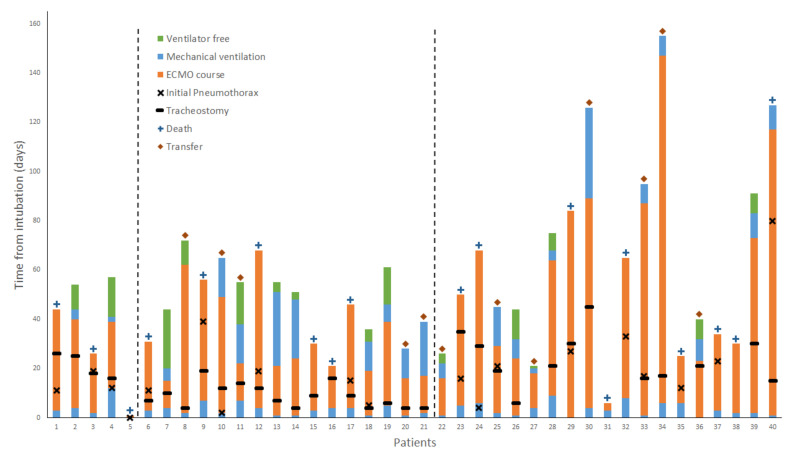
Hospital course of COVID-19 patients supported with veno-venous extracorporeal membrane oxygenation from the time of intubation. The diamond symbol indicates transfer out of our center, and, in the absence of a symbol above the bar, patients were discharged either home or to a rehabilitation facility. Patient 8 was on continuous trach collar, 4 days prior to ECMO decannulation. Patient 37 was extubated for 4 days while on ECMO, prior to tracheostomy. Patients before the first dashed line were cannulated in the period of “no tracheostomy” policy. Patients in between dashed lines were cannulated in the period of “early tracheostomy” policy. Patients after the second dashed line were cannulated in the period of “tracheostomy when no proven coagulopathy” policy.

**Figure 3 membranes-11-00306-f003:**
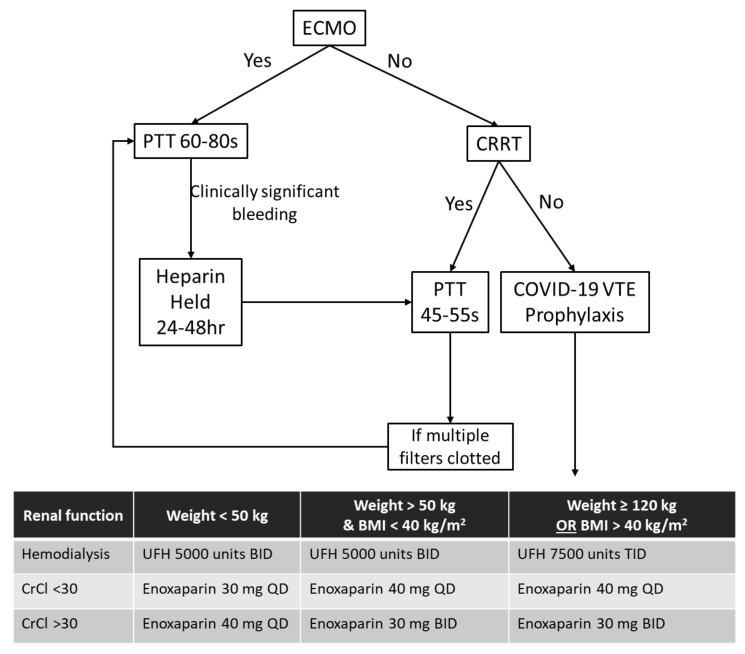
Anticoagulation algorithm for COVID-19 patients. Legend: BID—twice a day; CrCl—creatinine clearance; CRRT—continuous renal replacement therapy; ECMO—extracorporeal membrane oxygenation; PTT—partial thromboplastin time; QD—daily; TID—three times a day; and VTE—venous thromboembolism.

**Figure 4 membranes-11-00306-f004:**
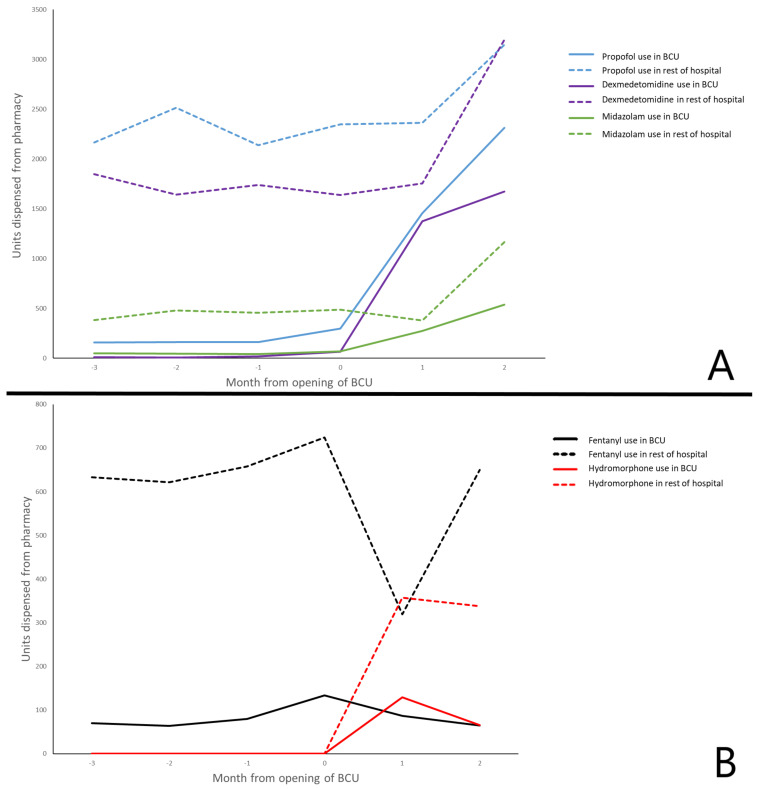
(**A**) Sedative use. (**B**) Narcotic use as sedation Legend: BCU—biocontainment unit.

**Table 1 membranes-11-00306-t001:** The criteria for veno-venous extracorporeal membrane oxygenation in COVID-19.

**Criteria for Consideration ***	**Contraindications ***
CO_2_ > 60 with pH < 7.25, or inadequate ventilation with P_plat_ lat0 wit_2_O	Age > 55 years BMI > 40 kg/m^2^
Ventilator/ms < 7.2	> 10 days on ventilator
PaO_2_/F_i_O_2_ ratio < 50 mm Hg with F_i_O_2_ > 80% for > 3 h *OR* a P_a_O_2_/F_i_O_2_ ratio < 80 mm Hg with F_i_O_2_ > 80% for > 6 h	Requirement for home O_2_ therapy for severe lung disease Severe neurological insult/injury
Despite optimization of mechanical ventilation (including APRV) *AND* despite attempts at rescue maneuvers (e.g., inhaled pulmonary vasodilators, prone positioning, neuromuscular blockade, etc.)	Terminal disease with low 1-year survival rate Multi-system failure WBC < 1000 cells/mL^3^ Poor baseline functional status
Consulting on-call team clinical discretion	

* Allows for discretion of on-call consulting ECMO team. Legend: APRV—airway pressure release ventilation; BMI—body mass index; and WBC—white blood cell.

**Table 2 membranes-11-00306-t002:** Patient characteristics and outcome.

**Variables**	Overall (n = 40)
Age (years)	43 (36, 50)
Sex (male)	33 (82.5)
MI (kg/m^2^)	34 (27, 40)
**Preexisting comorbidities**	
Asthma/COPD	3 (7.5)
Diabetes	12 (30)
**Pre-ECMO**	
Ventilator days	3 (1, 4)
pH	7.28 (7.19, 7.32)
CO_2_ (mm Hg)	62 (51.5, 76)
P/F ratio	69 (55, 78)
PIP (cm H_2_O)	38 (34, 42)
Mean airway pressure (cm H_2_O)	25 (23, 27)
Creatinine (mg/dL)	0.85 (0.65, 1.54)
Lactate (mmol/L)	2.2 (1.8, 3.0)
Bilirubin (mg/dL)	0.9 (0.6, 1.4)
CRRT/iHD pre	2 (5)
Cardiac arrest with ROSC	4 (10)
Glascow Coma Score	11 (11, 11)
RESP score	4 (2, 5)
Pneumothorax	19 (47.5)
Survival	21 (52.5)

Legend: ARDS—acute respiratory distress syndrome; BMI—body mass index; COPD—chronic obstructive pulmonary disease; CRRT—continuous renal replacement therapy; ECMO—extracorporeal membrane oxygenation; iHD—intermittent hemodialysis; P/F—PaO2/FiO2; PIP—peak inspiratory pressure; RESP—respiratory ECMO survival prediction; ROSC—return of spontaneous circulation.

## Data Availability

The datasets generated and/or analyzed during the current study are available from the corresponding author upon reasonable request.

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
