# Peer review of "A Dedicated Veno-Venous Extracorporeal Membrane Oxygenation Unit during a Respiratory Pandemic: Lessons Learned from COVID-19 Part II: Clinical Management"

_membranes, 2021, doi:10.3390/membranes11050306_

Round 1

Reviewer 1 Report

Shah and colleagues discuss the possibilities, advantages and disadvantages of a dedicated VV-ECMO unit for CARDS (COVID-19 ARDS). They argue that management of CARDS is different from ARDS after analyzing data from their center retrospectively. Forty CARDS patients requiring ECMO support were treated in a dedicated airlock biocontainment unit (BCU) in Baltimore, 21 of whom survived to discharge from BCU.

This is a rather long, but eloquently written article that offers limited novelty. Scientific impact is rather low, considering the small sample size, single-center retrospective design and lack of a control cohort. The description of the University of Maryland approach is however still interesting, because many other centers do not necessarily restrict VV-ECMO to a BCU.

Major concerns:

  1. I would urge to focus on the - interesting - question from the title: What can we learn from a dedicated BCU approach to manage patients with VV-ECMO in CARDS?
  2. This is a strictly descriptive report of single-center expert opinions: the article does not adhere to good scientific practice standards in its present form. It is a description of local procedures and experience, reflects expert opinion rather than scientific data.
  3. Materials and Methods provides insufficient information. A total of 7 lines (!) of methodology may be interpreted as a presumptuous request to the reader to simply believe that what happens in Baltimore is best practice. This is not acceptable. How were patients included? How were data acquired? How were treatment decisions made? How were data analyzed? etc. etc.
  4. The results section provides some information about treatment decision making (e.g. Fig 1). But what about established scores (RESP score, etc)?
  5. Why limit ECPR? Only because you thought that establishing proper PPE would take too long? Why describing this – the article focuses on VV-ECMO, not VA-ECMO and/or ECPR.
  6. Why no dual-lumen cannula?
  7. Why placing an arterial sheath? Just because you thought that it might be needed sometime?
  8. Most of the statements described in “Lessons” are not supported by data at all. E.g.: “Patients with CARDS rarely require VA-ECMO support and cardiac depression may 174 portend poor outcomes regardless of hybrid and VA-ECMO strategies.”. Please restrict to statements/conclusions that are supported by data.

Minor issue:

  1. page 2, line 71 and other locations: [Dave+Shah et al.] appears to be an unformatted citation

Reviewer 2 Report

The article describes the clinical management of covid-19 ECMO patients in a dedicated ECMO unit during the covid pandemic. The authors present the results of their first 40 patients and elaborate on different aspects of care with lessons learnt.

In the results section and in figure 3 the hospital course is shown. I would like to see a flowchart of patients (+/- pneumothorax) or a more standard table with baseline demographics including ecmo duration, hospital LOS. 

In figure 3 not all patients have transferred out of the BCU but we are now > 6 months further. Where are these patients?

3.2 line 174-175: would you support recommendation only using VV ECMO in CARDS en not give VA ECMO due to the poor prognosis?

3.3 right ventricular strain; please be more specific on the alterations from your normal practice.

Figure 2: please add desired saturation

Figure 2: Do you give BB in hypoxic patients? What do you think of the risk that BB may actually decrease DO2 despite an increase in measured arterial oxygen saturation. Do you meassure CO in these situations? serial lactate?

Line 263-266 although the optimal strategy might be unknown. Cannulation and ventilating every patient in the same way increases safety .

3.5 very interesting observation. What were the inspiratory pressures of these patients before and after starting ECMO? Dou you perform placement with or without heparin? is there a role for using bedside ultrasound?

Reviewer 3 Report

Shah and colleagues shared their experience in this interesting paper, offering insights and suggestions regardind management of patients in ECMO for COVID ARDS.

Round 2

Reviewer 1 Report

Thank you for this extensive revision. My main concerns have been addressed. Details have been added to Mat/Meth as one example (although this sections remains to be sub-optimal...). I appreciate that more patient data were amended (p4). I have no further issues.

Reviewer 2 Report

The article describing their experience in taking care of CARDS patients. The authors also point out that every hospital might be different. The response from the authors is detailed. My suggestions have been added to the content.